# Protocol for the cultural adaptation of pulmonary rehabilitation and subsequent testing in a randomised controlled feasibility trial for adults with chronic obstructive pulmonary disease in Sri Lanka

Akila R Jayamaha [iD],[1] Chamilya H Perera,[1] Mark W Orme [iD],[2,3] Amy V Jones,[2,3] Upendra K D C Wijayasiri,[4] Thamara D Amarasekara,[5] Ravini S Karunatillake,[6] Amitha Fernando,[6] Anthony L P Seneviratne,[7] Andy Barton,[8] Rupert Jones,[8] Zainab K Yusuf,[2,3] Ruhme B Miah,[2,3] Dominic Malcolm,[9] Jesse A Matheson,[10] Robert C Free,[3] Adrian Manise,[3] Michael C Steiner,[2,3] Savithri W Wimalasekera,[11] Sally J Singh[2,3]

For numbered affiliations see end of article.

**Correspondence to**
Akila R Jayamaha;
1226arj@gmail.com

## ABSTRACT

**Introduction** International guidelines recommend pulmonary rehabilitation (PR) should be offered to adults living with chronic obstructive pulmonary disease (COPD), but PR availability is limited in Sri Lanka. Culturally appropriate PR needs to be designed and implemented in Sri Lanka. The study aims to adapt PR to the Sri Lankan context and determine the feasibility of conducting a future trial of the adapted PR in Sri Lanka.

**Methods and analysis** Eligible participants will be identified and will be invited to take part in the randomised controlled feasibility trial, which will be conducted in Central Chest Clinic, Colombo, Sri Lanka. A total of 50 participants will be recruited (anticipated from April 2021) to the trial and randomised (1:1) into one of two groups; control group receiving usual care or the intervention group receiving adapted PR. The trial intervention is a Sri Lankan-specific PR programme, which will consist of 12 sessions of exercise and health education, delivered over 6 weeks. Focus groups with adults living with COPD, caregivers and nurses and in-depth interviews with doctors and physiotherapist will be conducted to inform the Sri Lankan specific PR adaptations. After completion of PR, routine measures in both groups will be assessed by a blinded assessor. The primary outcome measure is feasibility, including assessing eligibility, uptake and completion. Qualitative evaluation of the trial using focus groups with participants and in-depth interviews with PR deliverers will be conducted to further determine feasibility and acceptability of PR, as well as the ability to run a larger future trial.

**Ethics and dissemination** Ethical approval was obtained from the ethics review committee of Faculty of Medical Sciences, University of Sri Jayewardenepura, Sri Lanka and University of Leicester, UK. The results of the trial will be disseminated through patient and public involvement events, local and international conference proceedings, and peer-reviewed journals.

**Trial registration number** ISRCTN13367735

## Strengths and limitations of this study

► This study is the first examining the feasibility and acceptability of a culturally appropriate pulmonary rehabilitation programme for adults with chronic obstructive pulmonary disease in Sri Lanka.

► Taking a mixed-method approach, this study will provide a rich insight into delivering a trial of pulmonary rehabilitation in the Sri Lankan context.

► This study is a single-centre feasibility trial. As such, while findings will be an important first step in understanding the potential role of pulmonary rehabilitation in Sri Lanka and low-income and middle-income countries more broadly, findings may not be generalisable to other regions of the world.

## INTRODUCTION

Chronic obstructive pulmonary disease (COPD) is a progressive disease characterised by airflow obstruction and breathlessness. COPD is a major cause of morbidity and mortality throughout the world, corresponding to 6% of all deaths worldwide.[1] Further, more than 90% of COPD deaths occur in low-income and middle-income countries.[2] The most recent estimate of COPD prevalence in South Asia was 6.3%[3] with a prevalence in Sri Lanka of 10.5%,[4] similar to the estimated global COPD prevalence of

11.7%.[5] COPD is a significant burden to both patients and healthcare services.[6]

Chronic cough with sputum, breathlessness, physical inactivity and exercise intolerance resulting from dyspnoea or fatigue are common consequences of COPD.[1] Symptoms of COPD progressively worsen and people can become breathless, even at rest. Daily activities often become difficult as the condition worsens, impacting their quality of life.[7] The impact of COPD to the individual and to society makes the need for interventions to reverse the associated disability of paramount importance.

International guidelines recommend that pulmonary rehabilitation (PR) should be routinely offered to patients with chronic respiratory disease who have persistent symptoms, limited activity and/or are unable to adjust to illness.[8 9] It is a low-cost, high-impact intervention that improves the quality of life, reduces suffering, reduces mortality and reduces economic loss, relieves dyspnoea and fatigue, improves exercise capacity, improves psychological and emotional function, and enhances an individual's self-management of their condition.[1] Having realised the benefits of PR in COPD, Western countries have incorporated this as an important structural component of healthcare delivery services.[10] Implementation of PR based as practiced in Western countries, in Sri Lanka requires adaptation to the local health service, population and culture. Despite its effectiveness, there is a significant need to understand the feasibility of conducting PR in low-resource Sri Lankan healthcare setting and the acceptability of PR among Sri Lankan adults living with COPD and healthcare staff involved in its delivery. Uptake and completion of PR even in Western countries is an ongoing challenge.[11] The need to maximise appeal of PR to patients and referrers is a global issue and one that is likely to be specific to a given location and population. There remains an unmet need for PR in Sri Lanka. To be successful, PR must be not only evidence based, but also designed and implemented in a manner sensitive to the context in which it is being delivered, such as culture and geography. Therefore, the aim of this study is to devise an appropriate PR programme and then determine the feasibility and acceptability of this programme for adults living with COPD in Sri Lanka and assess the potential for a future trial of its effectiveness.

The objectives of the study are to:

1. Explore the needs and perceptions of adults living with COPD, their caregivers/family members and healthcare professionals to inform the adaptations required for a PR programme suitable for the Sri Lankan context.
2. Determine the feasibility of conducting adapted hospital-based PR for people living with COPD.
3. Assess the acceptability of the PR among Sri Lankan adults living with COPD and healthcare staff involved in its delivery.
4. Describe any changes in health of the adults living with COPD following completion of PR.
5. Assess the feasibility of a future trial and estimate the required sample size.

## METHODS AND ANALYSIS

### Study design and registration

The proposed study will be conducted in three phases. Phase 1: a qualitative study will be conducted to inform the adaptations required to make PR specific to the Sri Lankan context. Phase 2 is a single-blind randomised control feasibility trial. Adults living with COPD will be randomised (1:1) into adapted PR or usual care. Phase 3: qualitative evaluation of the trial to determine the feasibility and acceptability of PR deliverers and participants. The trial will be conducted, analysed and reported according to the Standard Protocol Items: Recommendations for Interventional Trials statement[12] and trial has been registered on the ISRCTN website. Study was started on ethical approval on 24 July 2020 for phase 1 of the study. The design of the study and flow of participant enrolment is presented in figure 1.

### Study setting

The study will be carried out at the Central Chest Clinic, Colombo, Sri Lanka, which is a leading healthcare facility that provides treatment for adults with respiratory disease. A room allowing confidential discussions will be used for conducting focus group discussions among adults living with COPD, their caregivers /family members, and nurses. Semistructured interviews will be conducted among doctors and physiotherapists in a given quite room at the Central Chest Clinic as convenient to them and without interfering with their routine work. The venue for PR will be a large room at the Central Chest Clinic.

### Participants

Key informants and suitable participants for phase 1 of the study will be identified by the researchers with the help of healthcare professionals involved in the treatment of COPD at Central Chest Clinic. Suitable participants will be purposively selected and informed verbally about the study by the researchers. After receiving a study information sheet, potential participants will be contacted to arrange an appointment, if they wish to take part. An opportunity to ask questions will be provided. If willing to take part in the study, they will be asked to provide written informed consent. Multireligious and multiethnic Sri Lankan cultural diversity and gender will be considered when recruiting to the study. Adults living with COPD aged ≥18 years and Medical Research Council (MRC) dyspnoea score grade 2 or higher, family member aged ≥18 years and looks after a patient with COPD and healthcare professionals who have more than 1-year experience of managing patients with COPD and working in the government healthcare system of the country will be eligible to participate in the phase 1 of the study. Participants who provide written informed consent will be enrolled in the study.

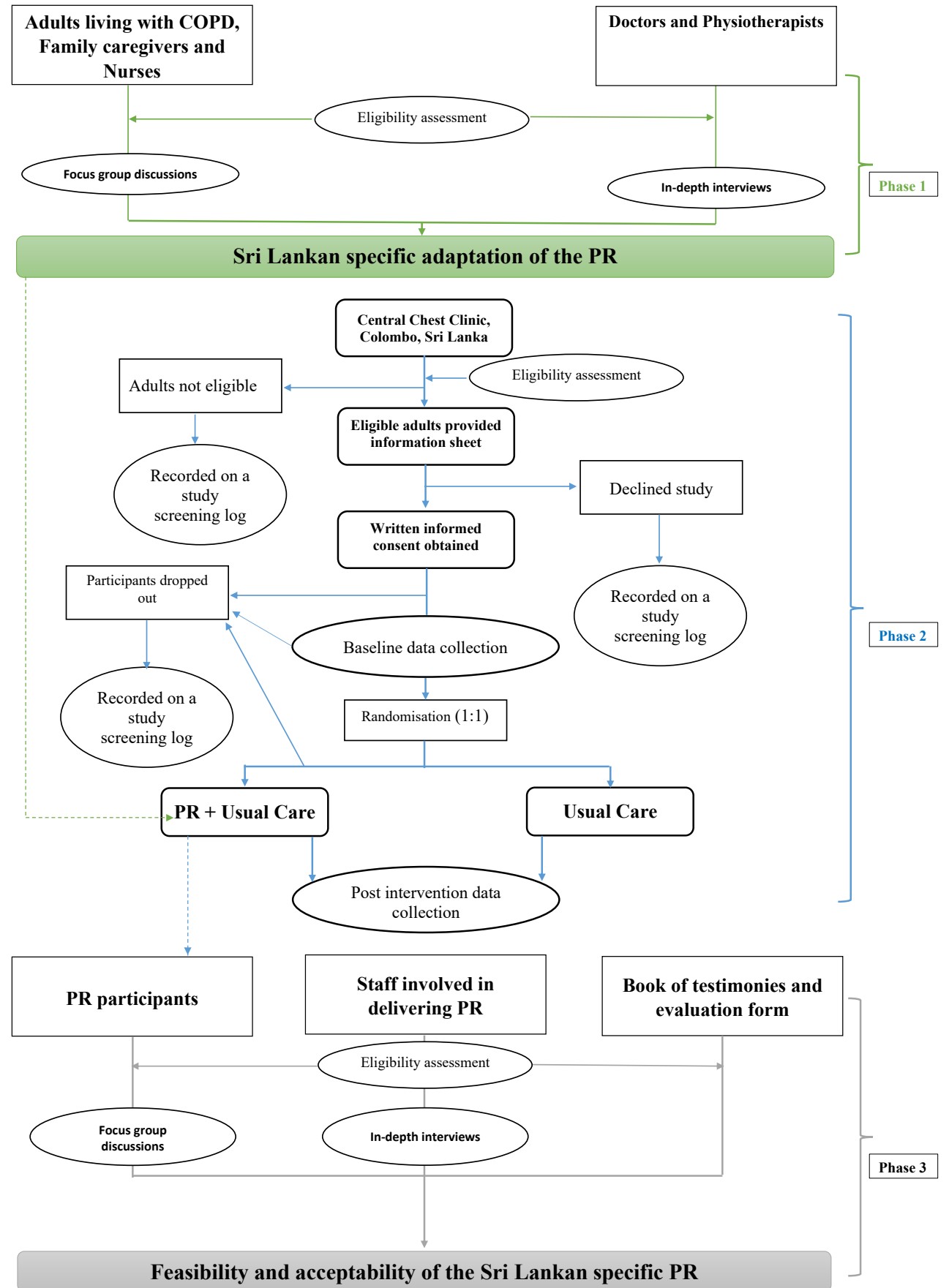

**Figure 1** Flow of participant of the three phases of the study. COPD, chronic obstructive pulmonary disease; PR, pulmonary rehabilitation.

After the adaptation of PR in phase 1, suitable participants for the phase 2 randomised controlled feasibility trial will be identified by the medical officers in the Chest Clinic and will be invited to take part in an eligibility assessment. This assessment will determine eligibility for entry into the study. People eligible for inclusion in the trial will be: aged ≥18 years, will have a clinically confirmed diagnosis of COPD by a physician, confirmed COPD with spirometry based on GOLD criteria with forced expiratory volume in the first second ($FEV_1$)/forced vital capacity (FVC)<0.7, and $FEV_1$ <80% predicted, ≥1 exacerbation required a hospitalisation in the year preceding study, mMRC grade ≥2 and willing to provide informed consent. Adults with comorbidities such as severe or unstable cardiovascular, other internal diseases and locomotor difficulties that preclude the exercise or malignant disease or other serious illness which will interfere with participation in the PR, will be excluded from the study. Individuals not eligible for the study will be recorded on a study screening log.

## Procedure

Eligible participants will be informed verbally about the study by the medical officers of the Central Chest Clinic. After receiving a study information sheet, potential participants will be contacted to arrange an appointment, if they wish to take part. An opportunity to ask questions will be provided. If willing to take part in the study, they will be asked to provide written informed consent. Information regarding the interest of participation in the study will be taken as field notes. Baseline outcome measures will be assessed. After baseline data collection, participants will be randomised (1:1) into two groups. One group will receive usual care and PR and the other group will receive only usual care which consists of pharmacological treatment and brief information about disease condition, medication and inhaler techniques. Experiences of the participants and PR deliverers regarding the acceptability and feasibility of PR will be explored in interviews and focus groups during phase 3 of the study. Participants who did not complete the PR will be asked to take part in a drop-out interview and information provided freely by the participants will be collected as field notes.

## Trial interventions
### Intervention: adapted PR

Sri Lankan-specific PR will comprise the core elements of an evidence-based rehabilitation, a programme of exercises[13 14] and health education.[13–15] The detail of delivery and adaptations of PR will be informed by findings of the phase 1 of the study. PR is typically a 6-week rolling programme that consists of 12 sessions in total.[13 14] Sessions usually last approximately 2 hours, with 1 hour for exercise training and 1 hour for education. All the education sessions and the instructions during PR will be provided in Sinhala language as convenient to the participants. The duration of the PR trial and frequency and duration of sessions will be discussed during phase 1. All staff delivering PR will be trained and assisted by a nursing officer, a clinical nutritionist, an exercise physician and a physiotherapist. The venue for PR will be at the Central Chest Clinic, Colombo, Sri Lanka. The Central Chest Clinic will have a maximum capacity of 10 patients per PR class. The equipment required will be simple and include chairs, weights and simple exercise equipment based on local availability, suitability and informed by qualitative work.[16]

In keeping with evidence-based guidelines,[13 14] discussions in phase 1 will include the core exercise component of PR, notably resistance training (eg, upper and lower limbs) and aerobic training (walking, sit-to-stand, step-ups) using minimal equipment and education. Each participant's exercise regimen will be individually prescribed and progressed[17] with walking speed will be prescribed using the Incremental Shuttle Walking Test (ISWT).[18] The process in which this is achieved by patients will be informed by phase 1. The education component will be delivered by an interdisciplinary team and will be informed by qualitative findings from phase 1 with discussions facilitated by usual education topics as described in guidelines.[13 14]

### Usual care

Usual care will consist of pharmacological treatment and optimisation, prescribed to patients after consultation with medical officers at the Central Chest Clinic. Brief information about disease condition, medication and inhaler techniques will be provided by medical officers and nurses. PR or any form of exercise recommendation is not part of usual care.

## Outcomes
### Primary outcome

The primary outcome of the trial will be the feasibility and the acceptability of the PR intervention.

#### Feasibility

Measures to assess feasibility are provided in table 1 and include the comprehensive assessment of the feasibility of patient recruitment and the intervention delivery.

#### Acceptability

The acceptability of the PR intervention among adults living with COPD and healthcare staff involved in its delivery will be assessed. Participants' experience of the PR, including any perceived benefits, challenges and changes they would make to the programme, will be explored in qualitative interviews and focus groups after their discharge assessment or withdrawal. The experience of healthcare professionals regarding the PR intervention, such as their confidence in delivering the programme, the components of PR, structure of PR, the patient adherence to the PR exercises and how their perceptions changed over the course of the trial, insights into barriers and facilitators to referral, uptake and completion of PR ((1) attending at least 10 out of 12 designated PR sessions and (2) attending the follow-up

**Table 1** Primary outcome measures—feasibility and operational experience assessment

| Feasibility of patient recruitment | Data sources |
|---|---|
| Feasibility of screening and recruiting participants | Interviews with the healthcare professionals, screening log |
| Suitability of the inclusion criteria | Interviews with the healthcare professionals, screening log |
| Number of eligible patients, number of patients screened, number of patients invited to take part, actual number of participants who consent to take part | Screening log |
| Number of patients who refuse, drop out and the reasons for refuse and drop out | Interviews with the patients, screening log |
| **Operational experience of intervention delivery** | |
| Service provider and multidisciplinary teams' willingness and ability to deliver the PR | Interviews with healthcare professionals |
| The practicality of delivering the intervention in the proposed setting | Interviews with healthcare professionals and focus groups with participants |
| The time needed to collect the data Baseline visit—time taken for each measure (each individual questionnaire and physical measure) Follow-up visit—time taken for each measure (each individual questionnaire and physical measure) | Interviews with the healthcare professionals, rehabilitation records |
| Data completeness and accuracy | Interviews with the healthcare professionals, rehabilitation records, RedCap |
| Adherence to home exercise | Interviews with the patients and self-report exercise diary |
| The training and resources needed to deliver the intervention (ensuring readiness for a future much larger multicentre trial) | Interviews with the healthcare professionals and focus groups with participants, rehabilitation records |
| Description of unintended events | Adverse events log, REDCap |

REDCap, Research Electronic Data Capture.

evaluation) will be explored in qualitative interviews at the end of the trial.

### Secondary outcomes
The secondary outcomes of this study are provided in table 2. Comparison of secondary outcome measures of baseline and post intervention will describe any changes in the health of the adults living with COPD following completion of PR.

### Sample size and recruitment target
This study is a feasibility trial that aims to provide data for an accurate estimation of the required sample size for future trials. Therefore, a formal sample size calculation is not required. We aim to recruit and randomise 50 participants to the study (25 in each group).

### Patient allocation, concealment and blinding
Randomisation will occur through random permuted blocks to either the intervention group (PR +usual care) or control group (usual care) in ratio 1:1 using Sealed envelope.

The computer-generated patient allocation sequence will be monitored by an individual who is independent of the research team and will inform the research team of group allocations via telephone. Participants will be informed about their group allocation after providing informed consent and completing baseline assessments.

It will not be possible to blind patients to their group allocation due to the nature of PR. Research staff will be blinded to outcome measures. Participants will be advised not to reveal their group during the follow-up assessment. Any episodes of unblinding will be documented and reported.

### Data collection
#### Phase 1: Qualitative assessment for adaptation of PR
*Focus groups with patient and family caregivers*
Focus groups with adults living with COPD and separate focus groups with their family members/caregivers will be conducted until data saturation. We anticipate conducting up to 5 focus groups with 6–8 participants in each. Data will be collected during patients' clinic visits. Focus group discussions will be audio recorded, expected to last approximately 45–90 min, and will be conducted face to face by an interviewer and note-taker (observer). Focus groups will be transcribed verbatim, with identifiable information removed. Envisaged outcomes of the focus groups will include Sri Lankan specific adaptation of PR.

*Interviews with healthcare professionals involving the treatment of COPD*
Up to 15 in-depth interviews with healthcare professionals will be conducted until data saturation. Structured

**Table 2** Secondary outcome measures

| Outcome measures | Baseline | Post intervention |
|---|---|---|
| Sociodemographics | x | |
| Lung health (spirometry data, smoking status, number of COPD exacerbations in the last year) | x | |
| Comorbidities | x | |
| Treatments | x | |
| Nutritional status (body mass index, mid upper arm circumference, skinfold thickness, self-report 7 days diet diary) | x | x |
| Disease burden (MRC dyspnoea grade, CAT, CCQ) | x | x |
| Economic impact of disease (WPAI) | x | x |
| Quality of life (EQ-5D-5L) | x | x |
| Pittsburgh Sleep Quality Index (PSQI) | x | x |
| Psychological well-being (Hospital Anxiety and Depression Scale) | x | x |
| Physical function (5× sit-to-stand test) | x | x |
| Exercise capacity (ISWT, ESWT) | x | x |
| International Physical Activity Questionnaire (IPAQ) | x | x |
| Accelerometer assessed physical activity (ActiGraph wGT3x-BT) | x | x |

CAT, COPD Assessment Test; CCQ, Clinical COPD Questionnaire; COPD, chronic obstructive pulmonary disease; EQ-5D-5L, EuroQol Five Dimensions Five Levels; ESWT, Endurance Shuttle Walk Tests; ISWT, Incremental Shuttle Walk Test; MRC, Medical Research Council; WPAI, Work Productivity and Activity Impairment questionnaire.

interviews will be conducted with healthcare professionals as convenient to them without interfering with their routine work. Interviews will be audio recorded and will be conducted face to face by an interviewer. Interviews will be transcribed verbatim, with identifiable information removed. Envisaged outcomes of the in-depth interviews will be a suitable design and content of PR acceptable respiratory healthcare deliverers in Sri Lanka.

### Phase 2: single-blind randomised control feasibility trial

Data will be collected by trained researchers, following standard operating procedures during participants' clinic visits. Baseline and post intervention assessments will be carried out by the blinded medical officers consulting at the study setting.

### Phase 3: qualitative evaluation of the PR intervention
*Focus groups with patients*

Participants allocated to the intervention group will be invited to participate in focus group discussions at the end of their PR programme. Focus groups will give an insight on views, experiences, opinions and recommendations which will inform future PR programmes. We anticipate conducting up to five focus groups until data saturation. Each focus group discussion will be conducted with 6–8 participants in each.

Focus group discussions will be audio recorded, expected to last approximately 45–90 min, and will be conducted by a trained moderator and a note-taker). Focus groups will be transcribed verbatim, with identifiable information removed. Consent will be obtained from participants prior to their involvement in focus groups.

*Interviews with PR staff*

Healthcare personnel involved in delivering PR will be invited to participate in in-depth interviews at the end of the study to discuss aspects of feasibility and acceptability, such as insights into barriers and facilitators to attendance, logistical barriers of running a PR programme and their perceptions, confidence of programme delivery and patients' experiences of the intervention. Details regarding previous experience on PR and prior training regarding PR will be assessed using brief questionnaire before commencing the in-depth interviews. We anticipate conducting up to 15 interviews, each expected to last approximately 15–60 min. Interviews will be audio recorded and will be conducted face to face by a trained interviewer. Interviews will be transcribed verbatim, with identifiable information removed.

*Book of testimonies and evaluation form*

Participants within PR will be asked to log their experience of PR as they progress through the programme. This will be in the form of a PR log book accessible to participants before, during and after sessions, as well as a dedicated evaluation form which will be provided as online supplemental material. Staff involved in PR will also receive the same evaluation form at the end of the study.

*Sample characteristics*

Basic demographics including age, sex, religion, nationality, marital status, age of leaving full-time education, education level, ethnicity, employment status, monthly income, lung health, smoking status (packs per year), biomass fuel exposure, primary respiratory diagnosis,

time since diagnosis in years, secondary respiratory diagnoses, family history of lung disease, comorbidities will be recorded at baseline.

## Lung function

Spirometry (post bronchodilator $FEV_1$, post bronchodilator FVC, $FEV_1$/FVC ratio), carboxyhaemoglobin test,[19] hospitalisations within the last 12 months, number of COPD exacerbations within the last 12 months, treatments will be collected as the baseline data prior to the randomisation of participants.

## Nutritional status

Height (Holtain stadiometer), weight (calibrated weight scale) will be measured and body mass index will be calculated. Mid upper arm circumference of the participants will be measured using a measuring tape and skinfold thickness of seven sites (triceps, mid axillary, subscapular, chest, abdominal, supra iliac and thigh measurements) of the body will be measured using a validated calliper. Triplicated measurements will be taken by the same investigator and mean value will be used for the calculation of body fat mass. Fat-free mass will be calculated using Durnin and Womersley formula.[20 21] Body composition analysis will be assessed using bioelectrical impedance (SFB7 Impedi Med dual frequency instrument, Impedi Med Limited, Australia) and software V.5.2.4.0.[22] Seven-day diet diary method will be used to assess the dietary history and calorie intake.

## Disease burden

Breathlessness will be measured using MRC dyspnoea grade (5-item).[23] The health-related quality of life related to mobility, self-care, usual activities, pain/discomfort and anxiety–depression will be measured using EuroQol Five Dimensions Five Levels (25-item).[24] Clinical status of the airways, functional limitations and psychosocial dysfunction will be determined using Clinical COPD Questionnaire (10-item).[25] COPD Assessment Test (8-item) will be used to determine the severity of the COPD.[26] Modified brief pain inventory (3-item) will determine the commonly reported chest pain; Work Productivity and Activity Impairment questionnaire (8-item) will be used to measure the effect of severity of COPD and general health on work productivity and regular activities[27] and Hospital Anxiety and Depression Scale (14-items) will be used to measure the patient's emotional state and the presence or absence of clinically significant anxiety and depression.[28] Sleep quality of the participants will be measured using Pittsburgh Sleep Quality Index.[29] Disease burden will be measured as baseline data prior to the randomisation and postintervention data.

## Exercise capacity

Following a safety assessment for absolute and relative contraindications for field walking tests,[30] maximal exercise capacity will be assessed using the ISWT[31] and Endurance Shuttle Walking Test (ESWT).[18] The ISWT and ESWT also include a pretest and post-test measure of $SpO_2$, blood pressure, Borg rating of breathlessness,[32] heart rate, Borg rate of perceived exertion.[33] The sit-to-stand test will be completed for five continuous repetitions, to measure lower limb movement and strength.[34] All measures of exercise capacity will be measured at baseline prior to the randomisation and post intervention.

## Physical activity

Subjective health-related physical activity will be measured with the International Physical Activity Questionnaire[35] at baseline prior to the randomisation and post intervention.

Participants will be asked to wear an activity monitor (ActiGraph wGT3X-BT, ActiGraph, Pensacola, Florida, USA), able to detect a range of PA intensities,[36] on the right anterior hip during waking hours for 7 days prior to attending PR (baseline) and for the 7 consecutive days prior to their discharge assessment (follow-up); taking it off for water-based activities and sleep. The collection of physical activity data will coincide with a 7 days diet diary. Printed instructions will be provided to the participants regarding how to wear the monitor.

Full accelerometry methodology is provided in table 3. Data will be processed as 60 s epoch files in ActiLife V.6.13.4 (ActiGraph). Non-wear will be defined as 60 min of consecutive zero counts with allowance for 2 min of interruptions.[37] Adherence to wearing the monitors will be assessed by examining the proportion of participants providing valid accelerometer data across a range of valid days (≥1–7 days) and minimum wear time (≥1–12 hours) thresholds. For reported group averages, only participants providing ≥4 valid days of ≥8 hours, for both time points will be included in the analyses.[38] Step count, time spent in different absolute intensity classifications of physical behaviours[39] and average movement intensity (activity counts per minute) will be reported. Time spent in physical activity matching participants' prescribed walking exercise intensity will be derived by aligning the average walking speed during the ESWT with the activity monitor counts per minute and cadence.

## Patient and public involvement

Adults living with COPD often tell us how having COPD impacts their lives and that they often find it challenging to know what they can do to better manage their condition. When they talk about their self-management strategies, it is clear that education and exercise support would be of value. Also, they revealed the necessity of a programme to support their condition and willingness to attend such a programme. Patients are generally positive about being able to access support but there is little available to them. The study was planned to fulfil the need of adults living with COPD. The delivery and adaptations of the trial intervention (Sri Lankan-specific PR) will also be informed by the adults living with COPD, their caregivers/family members and healthcare professionals. Priorities, experience and preferences of the stakeholders will be used to design the PR. Healthcare professionals involved in the treatment of COPD at Central Chest Clinic will

| Table 3 | Accelerometry data collection and processing parameters |
|---|---|
| Accelerometer Model | ActiGraph wGT3X-BT (V.6.13.4; firmware V.1.9.2) |
| Serial number range | Twenty unique devices will be used ranging from MOS2E09190601 to MOS2E24190146 averaging six deployments per device (same serial used for baseline and follow-up wear periods) |
| Piezosensor orientation | Triaxial |
| Mode setup | Mode 29 (x, y, z, steps, lux) |
| Original sample rate | 100 Hz (.gt3x file format) |
| Deployment method | Baseline:<br>Fitted by research team on day 0 (baseline PR assessment)<br>Fitted by participant on day 1<br>Follow-up:<br>Fitted by research team day 0 (11th/12th session)<br>Fitted by participant on day 1 |
| Location worn | Anterior hip adjacent to the mid-line of the thigh |
| Requested days of wear | 7 days of free living (days 1–7; 10 080 epochs) |
| Initialisation | Not deployed in delay mode in order to standardised capture of day 0 (00:00) with stop time based on date of first PR class (baseline) and data of discharge assessment (follow-up) |
| Wear instructions | Wear continuously except for sleep and water-based activities |
| Non-wear appropriation | ≥60 min of consecutive 0 s with allowance for 2 min of interruptions were deemed biologically implausible and coded as non-wear |
| Valid day criteria | ≥8 hours of valid waking wear time |
| Valid file | ≥4 valid days for each of the two time points |
| Missing data | Data modelling or imputation will not be performed |
| Epoch length | 60 s |
| Intensity classification (absolute) | Uniaxial (x-axis) cut-points as follows: sedentary time <100 cpm; light intensity 100–2019 cpm; moderate intensity 2020–5998 cpm; vigorous intensity ≥5999 cpm (moderate-to-vigorous intensity ≥2020 cpm) |
| Intensity classification (relative) | Uniaxial (x-axis) cut-points based on Endurance Shuttle Walk Test performance |

PR, pulmonary rehabilitation.

be involved in the recruitment as key informants and conduct of the study. The feasibility and acceptability of the PR intervention among adults living with COPD and healthcare staff involved in its delivery will be assessed in qualitative interviews at the end of the trial. The results of the trial will be disseminated through patient and public involvement events, local and international conference proceedings. As well as all the research participants, stakeholders and individuals with COPD will be openly invited to take part in an event organised at the central chest clinic to reveal the study findings.

### Costing

As part of this trial, the cost of creating a Sri Lankan-adapted PR programme will be calculated. This will include both single and recurrent costs. Single payments include the necessary costs to set up and run a programme, while recurrent cost refers to any item with a life expectancy of 1 year or less and typically includes disposable materials.[40] The fixed costs will be captured prior to the first participant enrolling into the programme and the recurrent costs will be collected at the mid-stage of recruitment. An average cost per participant will be calculated. Table 4 demonstrates the variables that will be used to calculate fixed and recurrent costs.

### Data management

Data collected during the study will be entered into a database using Research Electronic Data Capture (REDCap), which is a web-based platform.[41] Access to the database will be via a secure password-protected web interface. The participants will be identified by a study-specific identification code. Data will be validated using real-time data entry validation and electronic checks lead by the Independent Data Monitoring Committee (IDMC), established at the University of Leicester, UK.

### Quantitative data analysis

The data will be analysed using IBM SPSS Statistics for Windows, V.23.0. Data for baseline and follow-up timepoints will be presented as descriptive statistics as appropriate. No inferential statistics will be performed due to the feasibility design of the trial.

**Table 4** The variables used to calculate fixed and recurrent costs (not an exhaustive list)

| Fixed costs | Recurrent costs |
| --- | --- |
| Venue hire | Venue hire |
| Electrical equipment (laptop, printer, projector) | Staff time to conduct PR (assessment at baseline and discharge, conduct PR classes, telephone calls and data entry) |
| Equipment for PR (weights, treadmill, cycle ergometer, country-specific equipment, step-up box, chairs) | Disposable equipment (for blood glucose monitor, spirometer mouthpieces, nose-clips, glyceryl trinitrate spray) |
| Equipment for shuttle walking tests (cones, licences, stop watches, tape measure, electrical equipment to play audio) | Servicing costs (spirometer, PR equipment, specifically treadmills and cycle ergometers) |
| Equipment for PR assessment (height stadiometer, weight scales, sphygmomanometer, pulse oximeter, spirometer, calibration syringe, country-specific equipment) | Miscellaneous (oxygen cylinders, questionnaire licences, stationery (paper)) |
| Additional safety equipment (blood glucose monitor, oxygen cylinder holder) | |
| Miscellaneous (filing cabinets, storage units, questionnaire translations, questionnaire licences, staff uniform) | |

PR, pulmonary rehabilitation.

## Qualitative data analysis

Qualitative data will be analysed using thematic analysis. This approach follows six distinct stages: familiarisation with data; generating initial codes; searching for themes; reviewing themes; defining and naming themes and producing the report.[42] The responsible investigator will carry out initial coding and a sample of focus group transcripts will be coded by a second member of the team to improve consistency and to enhance interpretive authenticity. Throughout the data analysis, the team will meet to discuss and review emerging themes and search the accounts that provide contesting views of the same phenomena. Close attention will be paid to the complexity and interactions inherent in the focus group data.[43]

## Adverse events

All adverse events and serious adverse events will be recorded on an adverse event log, within study trial management paperwork, case report forms and REDCap. There will be no formal interim analysis of data due to the feasibility nature of the trial. The IDMC will review high-level safety data. Adverse events will be monitored at least every month, and as needed on an ad hoc basis, to ensure the continuing safety of the participants. The scientific committee will determine the need to terminate the trial. Participants who experience any such event will be directed to the appropriate hospital and all the necessary care will be ensured and followed up until the participant has resolved or stabilised.

## ETHICS AND DISSEMINATION

Ethical approval was obtained from the ethics review committee of the Faculty of Medical Sciences, University of Sri Jayewardenepura (FMS/USJP ERC 64/19) and the University of Leicester, UK (26770). Further permission for the proposed study will be obtained from the relevant authority at the Central Chest Clinic, Colombo. Sri Lanka. Privacy and the confidentiality of all information and identities of participants will be strictly maintained and will not be disclosed when publishing the results of the study. A copy of the test results obtained will be provided to each participant at the end of the trial for subsequent follow-up and treatment at the clinic. Compensation for travelling will be provided to all the participants. All study documents will be translated to Sinhala and Tamil to ensure clear communication. Participation will be without compulsion and each participant has the right to withdraw at any time, without providing a reason. Consent form and data sheets will be securely stored in a separate locked cupboard. Study computers will be password protected. The participants (adults living with COPD) of the phase 1 study also will be provided the opportunity to participate the PR. All the data will be stored safely up to 6 years and after 6 years consent form and data sheets will be disposed of appropriately. Study team and IDMC only will have access to final trial dataset. Data from the Global Health Research Group on Respiratory Rehabilitation (Global RECHARGE) Core Dataset [44] will be made available following the completion of this project and we are considering the best tools to use to make this database available to the wider community. Any modifications of the protocol will be updated on trial registry (ISRCTN) and will be informed to the ethics review committee and the participants. Participants will be provided the provisions for reconsenting after any change of the approved protocol. It will be made clear in the publication of trial findings.

**Author affiliations**
[1]Health Sciences, KIU, Colombo, Sri Lanka
[2]Centre for Exercise and Rehabilitation Science, NIHR Leicester Biomedical Research Centre-Respiratory, University Hospitals of Leicester NHS Trust, Leicester, UK
[3]Department of Respiratory Sciences, University of Leicester, Leicester, UK
[4]Sports Medicine Unit, Colombo South Teaching Hospital, Kalubowila-Dehiwela, Sri Lanka
[5]Faculty of Allied Health Sciences, University of Sri Jayewardenepura, Nugegoda, Sri Lanka
[6]Central Chest Clinic, National Hospital of Sri Lanka, Colombo, Sri Lanka
[7]Primary Care Respiratory Group, Colombo, Sri Lanka
[8]Faculty of Health, University of Plymouth, Plymouth, UK
[9]School of Sport, Exercise and Health Sciences, Loughborough University, Loughborough, UK
[10]Department of Economics, University of Sheffield, Sheffield, UK
[11]Faculty of Medical Sciences, University of Sri Jayewardenepura, Nugegoda, Sri Lanka

**Contributors** All authors have substantially contributed to the conception and design of the study. ARJ drafted the manuscript. All authors of the paper have revised the content and approved the final version to be published. All authors are accountable for all aspects of the work.

**Funding** This research was funded by the National Institute for Health Research (NIHR) (17/63/20) using UK aid from the UK Government to support global health research.

**Disclaimer** The views expressed in this publication are those of the author(s) and not necessarily those of the NIHR or the UK Department of Health and Social Care.

**Competing interests** None declared.

**Patient consent for publication** Not required.

**Provenance and peer review** Not commissioned; externally peer reviewed.

**ORCID iDs**
Akila R Jayamaha http://orcid.org/0000-0002-3372-4537
Mark W Orme http://orcid.org/0000-0003-4678-6574

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
