## [Reviewer comments · BMJ Open]

ARTICLE DETAILS

TITLE (PROVISIONAL)	Protocol for the Cultural Adaptation of Pulmonary Rehabilitation and Subsequent Testing in a Randomised Controlled Feasibility Trial for Adults with Chronic Obstructive Pulmonary Disease in Sri Lanka
AUTHORS	Jayamaha, Akila; Perera, Chamilya; Orme, Mark; Jones, Amy; Wijayasiri, Upendra; Amarasekara, Thamara; Karunatillake, Ravini; Fernando, Amitha; Seneviratne, Anthony; Barton, Andy; Jones, Rupert; Yusuf, Zainab; Miah, Ruhme; Malcolm, Dominic; Matheson, Jesse; Free, Robert; Manise, Adrian; Steiner, michael; Wimalasekera, Savithri; Singh, Sally

VERSION 1 – REVIEW

REVIEWER	Claire Nolan Royal Brompton and Harefield NHS Foundation Trust
REVIEW RETURNED	01-Sep-2020

GENERAL COMMENTS	Jayamaha and colleagues plan to undertake a feasibility trial to explore the cultural adaptation of a pulmonary rehabilitation programme for patients with COPD in Sri Lanka. The rationale and methodology is appropriate and relevant. Minor comment: 1. The authors need to add the study dates as per BMJ Open guidance for protocols. 2. The study documents have been provided in English only. Given the purpose of the study is to develop a culturally appropriate intervention, it would be informative to describe what language the study will be conducted in and why e.g. English, Tamul, Sinhala. Will this influence the questionnaire-based outcome measures?
---

REVIEWER	Narelle Cox Monash University, Australia
REVIEW RETURNED	22-Sep-2020

GENERAL COMMENTS	Thank you for the opportunity to review this very interesting study protocol, that will work toward improving access to PR in a presently underserved location. I have a few queries for the authors to consider: Could the authors confirm that ethics approval has not yet been obtained and recruitment has not yet started? (as indicated) or update accordingly if this has changed. (Trial registration for Phase I indicates ethics approval has been obtained for this)
---

	Comparison to trial registration: 1. The trial registration for Phase I indicates this will comprise a survey of some 408 participants, whereas the manuscript indicates Phase I will be qualitative interviews of approx. 40 patients, 40 care-givers and 15 healthcare professionals. Might the authors want to comment on these difference/changes? Or, on reflection, is it that the provided trial registration for Phase I doesn't actually apply to this protocol manuscript; and rather that the current protocol has 3 components – i) qualitative interviews ii) feasibility RCT and iii) followup qualitative interviews and thus relates only to trial registration for Phase II & III? If this is the case, should the reference to Trial Protocol ISRCTN58273367 be removed? Alternatively, could the authors consider a section in the manuscript that details any changes from the registration? 2. Trial registration ISRCTN13367735 indicates up to 40 people with COPD will be recruited. In the main text this appears to apply to Phase I qualitative interviews only, with 60 participants being recruited for the RCT. Do the authors wish to address this? Abstract – 3. Might it be clearer to indicate that the 12 sessions are delivered over 6 weeks? Main text – 4. Page 3, Line 30-31 'the development of rehabilitation.....will ensure Universal Healthcare Coverage'. It is unclear the meaning/context of this sentence. Can this sentence be amended or removed? 5. Page 5 Study setting It is unclear why only 'nurses' are mentioned relative to focus groups of patients and care-givers. Is it necessary to delineate the location/room where interviews will occur? Participants 6. Could the authors please provide more detail on 'how' participants will be identified eg. a register of COPD patients? referrals? outpatient clinic attendance? for Phase I. 7. Inclusion criteria for Phase II (RCT) indicate at least one exacerbation per year. Is this an exacerbation requiring a hospitalisation? Is it specifically in the year preceding study enrolment? Additional clarification would be useful. 8. Figure 1 Can the authors please provide detail in the text as to what comprises the 'eligibility assessment' for Phase I. Relating to Phase II – is the green dotted line/arrow to PR +usual care required? Its meaning isn't clear. Baseline data collection is indicated twice during Phase II. Amend second listing to 'post-intervention' or similar. If there are additional details beyond 'participating in Phase II' that comprise eligibility assessment for Phase III, it would be useful if these were documented in the text. Procedure
--	--

	9. This may be a formatting requirement of the journal, but I was wondering if details of the processes as are intended in relation to Phase I should be detailed here also? Trial interventions (page 8) 10. Wording in the first paragraph on page 8 is unclear. Is it that the PR program will be in keeping with guidelines and discussions from Phase I? or is this text intending to describe the question process to be used in Phase I? Outcome measures 11. Would there be any merit in collection of data relating to any staff training eg. number of staff; disciplines; type of training; duration – to add to feasibility outcomes? Likewise, to better detect change in staff perceptions/confidence/barriers to delivery might pre- and post- staff interviews relative to the RCT be useful? 12. The authors have rightly indicated the challenges of both uptake and completion of PR globally. Uptake and completion are both listed as part of the primary outcomes for feasibility in the trial registration, but not indicated or defined in the manuscript. If planning to use to demonstrate feasibility could the authors to provide their definitions for uptake and completion of PR in this protocol?
--	---

VERSION 1 – AUTHOR RESPONSE

Reviewer(s)' Comments to Author:

Reviewer: 1

Reviewer Name: Claire Nolan

Institution and Country: Royal Brompton and Harefield NHS Foundation Trust

Please state any competing interests or state 'None declared': None declared

Please leave your comments for the authors below

Jayamaha and colleagues plan to undertake a feasibility trial to explore the cultural adaptation of a pulmonary rehabilitation programme for patients with COPD in Sri Lanka. The rationale and methodology is appropriate and relevant.

Minor comment:

1. The authors need to add the study dates as per BMJ Open guidance for protocols.

Study was commenced upon ethical approval on 24/07/2020 for Phase 1 of the study (line no: 101 - 102, page no: 4)

2. The study documents have been provided in English only. Given the purpose of the study is to develop a culturally appropriate intervention, it would be informative to describe what language the study will be conducted in and why e.g. English, Tamul, Sinhala. Will this influence the questionnaire-based outcome measures?

We agree that, describing what language the study will be conducted in and why, is important and have amended the text accordingly.

All the education sessions and the instructions during PR will be provided in Sinhala language as convenient to the participants. (line no: 161 -163, page no: 7)

It is mentioned in the “ethics and dissemination” section that, all study documents will be translated to Sinhala and Tamil to ensure clear communication. (line no: 409 - 410, page no: 18)

Reviewer: 2

Reviewer Name: Narelle Cox

Institution and Country: Monash University, Australia

Please state any competing interests or state 'None declared': None declared

Please leave your comments for the authors below

Thank you for the opportunity to review this very interesting study protocol, that will work toward improving access to PR in a presently underserved location.

I have a few queries for the authors to consider:

Could the authors confirm that ethics approval has not yet been obtained and recruitment has not yet started? (as indicated) or update accordingly if this has changed. (Trial registration for Phase I indicates ethics approval has been obtained for this)

Study was commenced upon ethical approval on 24/07/2020 for Phase 1 of the study. (line no: 101 - 102, page no: 4)

Comparison to trial registration:

1. The trial registration for Phase I indicates this will comprise a survey of some 408 participants, whereas the manuscript indicates Phase I will be qualitative interviews of approx. 40 patients, 40 care-givers and 15 healthcare professionals. Might the authors want to comment on these difference/changes? Or, on reflection, is it that the provided trial registration for Phase I doesn't actually apply to this protocol manuscript; and rather that the current protocol has 3 components – i) qualitative interviews ii) feasibility RCT and iii) followup qualitative interviews and thus relates only to trial registration for Phase II & III? If this is the case, should the reference to Trial Protocol ISRCTN58273367 be removed? Alternatively, could the authors consider a section in the manuscript that details any changes from the registration?

Trial Protocol ISRCTN58273367 was provided in error and has been removed as suggested. We apologise for this error and the confusion caused. (line no: 34, page no: 2)

2. Trial registration ISRCTN13367735 indicates up to 40 people with COPD will be recruited. In the main text this appears to apply to Phase I qualitative interviews only, with 60 participants being recruited for the RCT. Do the authors wish to address this?

We apologise for the confusion and have updated the manuscript to reflect our aim of recruiting 50 patients (25 in each arm) to the main trial (Stage II). We noticed that 60 participants was written in error. (line no: 21, page no: 1)

Up to 40 patients will be recruited as part of the qualitative work in Phase I (e.g. 5 FGDs of 8 patients).

Abstract –

3. Might it be clearer to indicate that the 12 sessions are delivered over 6 weeks?

Corrected as suggested "The PR programme is likely to consist of 12 sessions of exercises and health education, delivered over 6 weeks." (line no: 23 - 2, page no: 1)

Main text –

4. Page 3, Line 30-31 'the development of rehabilitation.....will ensure Universal Healthcare Coverage'. It is unclear the meaning/context of this sentence. Can this sentence be amended or removed?

Sentence was removed as suggested

5. Page 5 Study setting

It is unclear why only 'nurses' are mentioned relative to focus groups of patients and care-givers. Is it necessary to delineate the location/room where interviews will occur?

Nurses will be the main referrers to the trial and to PR (if ultimately provided as a clinical service) whereas Doctors and Physiotherapists will be involved in the delivery of PR classes and service provision. It is also more feasible to arrange Nurses to attend focus group discussions, flexibility we do not have with Doctors and Physiotherapists. Therefore, the decision was made to run focus groups for nurses and interviews for other healthcare workers.

Unnecessary details regarding the location were removed and we have modified the content as below:

"Semi structured interviews will be conducted among doctors and physiotherapists in the conference hall or consultation rooms as convenient to them and without interfering with their routine work." (line no: 108 - 110, page no: 5)

Participants

6. Could the authors please provide more detail on 'how' participants will be identified eg. a register of COPD patients? referrals? outpatient clinic attendance? for Phase I.

Key informants and suitable participants for phase 1 of the study will be identified by the researchers with the help of health care professionals involved in the treatment of COPD at Central Chest Clinic. Suitable participants will be purposively selected and informed verbally about the study by the researchers. After receiving a study information sheet, potential participants will be contacted to arrange an appointment, if they wish to take part. An opportunity to ask questions will be provided. If willing to take part in the study, they will be asked to provide written informed consent. (line no: 113 - 119, page no: 5)

7. Inclusion criteria for Phase II (RCT) indicate at least one exacerbation per year. Is this an exacerbation requiring a hospitalisation? Is it specifically in the year preceding study enrolment? Additional clarification would be useful.

Additional clarifications added as suggested

"≥1 exacerbation required a hospitalisation in the year preceding study" (line no: 131 - 132, page no: 5)

8. Figure 1

Can the authors please provide detail in the text as to what comprises the 'eligibility assessment' for Phase I.

The 'eligibility assessment' for Phase 1 was included as follows:

"Adults living with COPD aged ≥18 years and Medical Research Council (MRC) dyspnoea score grade 2 or higher, Family member aged ≥18 years and looks after a patient with COPD and Health care professionals who have more than 1-year experience of managing patients with COPD and working in the government health care system of the country will be eligible to participate in the phase 1 of the study." (line no: 120 - 125, page no: 5)

Relating to Phase II – is the green dotted line/arrow to PR +usual care required? Its meaning isn't clear.

The green dotted line/arrow indicate that, “the participants (adults living with COPD) of the phase 1 study also will be provided the opportunity to participate the PR.” (figure 01, page 7)

The schematic diagram has been edited to improve clarity.

Baseline data collection is indicated twice during Phase II. Amend second listing to ‘post-intervention’ or similar.

Corrected as the “post intervention data collection” (figure 01, page 6)

If there are additional details beyond ‘participating in Phase II’ that comprise eligibility assessment for Phase III, it would be useful if these were documented in the text.

Same eligibility criteria will be use to recruit the Phase III qualitative assessment as we are only asking participants who have enrolled to the main trial (Phase II) to participate in Phase III.

Procedure

9. This may be a formatting requirement of the journal, but I was wondering if details of the processes as are intended in relation to Phase I should be detailed here also?

We have formatted according to the journal’s requirements therefore have not made changes to the content of these sections. We hope that having these as consecutive sections will help the reader when the paper is formatted to the journal’s style (e.g. the figure will not fall between them).

Trial interventions (page 8)

10. Wording in the first paragraph on page 8 is unclear. Is it that the PR program will be in keeping with guidelines and discussions from Phase I? or is this text intending to describe the question process to be used in Phase I?

Reworded the first paragraph on page 8 as:

“Sri-Lankan specific PR will comprise the core elements of an evidence based rehabilitation, a programme of exercises and health education in keeping with guidelines. The detail of delivery and adaptations of PR will be informed by the findings of the Phase 1 of the study.” (line no: 157 - 159, page no: 6)

Outcome measures

11. Would there be any merit in collection of data relating to any staff training eg. number of staff; disciplines; type of training; duration – to add to feasibility outcomes? Likewise, to better detect change in staff perceptions/confidence/barriers to delivery might pre- and post- staff interviews relative to the RCT be useful?

We agree that, assessing the perceptions/confidence/barriers and the training will be interesting and we amended the brief assessment as suggested.

“Health care personnel involved in delivering PR will be invited to participate in in-depth interviews at the end of the study to discuss aspects of feasibility and acceptability, such as insights into barriers and facilitators to attendance, logistical barriers of running a PR programme their perceptions, confidence of programme delivery and patients’ experiences of the intervention. Details regarding previous experience on PR and prior training regarding PR will be assess using brief questionnaire before commencing the in-depth interviews”. (line no: 260 - 265, page no: 11, 12)

12. The authors have rightly indicated the challenges of both uptake and completion of PR globally. Uptake and completion are both listed as part of the primary outcomes for feasibility in the trial registration, but not indicated or defined in the manuscript. If planning to use to demonstrate feasibility could the authors to provide their definitions for uptake and completion of PR in this protocol?

Added the definition for completion of PR in this protocol as follows:

“The experience of healthcare professionals regarding the PR intervention, such as their confidence in delivering the programme, the components of PR, structure of PR, the patient adherence to the PR exercises and how their perceptions changed over the course of the trial, insights into barriers and facilitators to referral, uptake and completion of PR ((i) attending at least 10 out of 12 designated PR sessions and (ii) attending the follow-up evaluation) will be explored in qualitative interviews at the end of the trial” (line no: 197 - 203, page no: 9)

FORMATTING AMENDMENTS (if any)

Required amendments will be listed here; please include these changes in your revised version:

1. Required figure/s format:

- Figures can be supplied in TIFF, JPG or PDF format (figures in document, excel or powerpoint format will not be accepted), we also request that they have a resolution of at least 300 dpi and 90mm x 90mm of width. Please see the following link for further details on preparing images for submission: <https://authors.bmj.com/writing-and-formatting/formatting-your-paper/>

We have changed the format of the figure as suggested by the journal guidelines. (Figure 01)

2. Required Supplementary format:

- Please re-upload your Supplementary files in PDF format.

All the supplementary files converted in to PDF format and uploaded as suggested.

3. Same figure embedded:

- Upon checking your manuscript files, you've already uploaded the same figure embedded on your main document. Kindly delete the same figures embedded on your main document. Please note that we don't accept figures embedded on main document file.

We apologise for the inconvenience and deleted the figure embedded in the main document. (line no: 137, page no: 6) Figure 01 is attached as a supplementary file in PDF format.

4. Patient and Public Involvement:

- We have implemented an additional requirement to all articles to include 'Patient and Public Involvement' statement within the main text of your main document. Please refer below for more information regarding this new instruction:

Authors must include a statement in the methods section of the manuscript under the sub-heading 'Patient and Public Involvement'.

This should provide a brief response to the following questions:

How was the development of the research question and outcome measures informed by patients' priorities, experience, and preferences?

How did you involve patients in the design of this study?

Were patients involved in the recruitment to and conduct of the study?

How will the results be disseminated to study participants?

For randomised controlled trials, was the burden of the intervention assessed by patients themselves?

Patient advisers should also be thanked in the contributorship statement/acknowledgements.

If patients and or public were not involved please state this.

Patient and Public Involvement

Adults living with COPD often tell us how having COPD impacts their lives and that they often find it challenging to know what they can do to better manage their condition. When they talk about their self-management strategies, it is clear that education and exercise support would be of value. Also,

they revealed the necessity of a programme to support their condition and willingness to attend such a programme. Patients are generally positive about being able to access support but there is little available to them. The study was planned to fulfill the need of adults living with COPD. The delivery and adaptations of the trial intervention (Sri-Lankan specific PR) will also be informed by the adults living with COPD, their care-givers /family members and health care professionals. Priorities, experience, and preferences of the stakeholders will be utilized to design the PR. Health care professionals involved in the treatment of COPD at Central Chest Clinic will be involved in the recruitment as key informants and conduct of the study. The feasibility and acceptability of the PR intervention among adults living with COPD and healthcare staff involved in its delivery will be assessed in qualitative interviews at the end of the trial. The results of the trial will be disseminated through patient and public involvement events, local and international conference proceedings. As well as all the research participants, stakeholders and individuals with COPD will be openly invited to take part in an event organize at the central chest clinic to reveal the study findings.

(line no: 342 - 359, page no: 15, 16)

5. Contributorship statement format:

- Please provide a more detailed contributorship statement. It needs to mention all the names/initials of authors along with their specific contribution/participation for the article.

This should list each author's contribution to the paper according to the ICMJE guidelines for authorship. This should be stating how each author contributed to the article. It should discuss on the planning, conduct and reporting of the work in your paper. You may also consider the conception and design, acquisition of data or analysis and interpretation of data, etc.

Contributorship statement is elaborated as suggested by adding the

Author contribution

All authors (Akila R Jayamaha, Chamilya H Perera, Mark W Orme, Amy V Jones, Upendra K D C Wijayasiri, Thamara D Amarasekara, Ravini de S Karunatilake, Amitha C Fernando, Seneviratne A L P De S., Andy Barton, Rupert Jones, Zainab Yusuf, Ruhme B Miah, Dominic Malcolm, Jesse A Matheson, Robert C Free, Adrian Manise, Michael C Steiner, Savithri W Wimalasekera, Sally J Singh) have substantially contributed to the conception and design of the study. Akila R Jayamaha drafted the manuscript. All authors of the paper have revised the content and approved the final version to be published. All authors are accountable for all aspects of the work. (line no: 440 - 448, page no: 19, 20)

VERSION 2 – REVIEW

REVIEWER	Narelle Cox Monash University
REVIEW RETURNED	Narelle Cox Monash University
GENERAL COMMENTS	The authors have made numerous amendments to their manuscript that have improved its clarity. Good luck with your study! (Please note, I could not see Figure in the re-submitted documents but have read the authors description of changes.

VERSION 2 – AUTHOR RESPONSE

Reviewer(s)' Comments to Author:

Reviewer: 2

Narelle Cox

Monash University

Please state any competing interests or state 'None declared':

None declared

Competing interests was declared as suggested,

Competing interests None declared. (line no: 450, page no: 20)

Comments to the Author

The authors have made numerous amendments to their manuscript that have improved its clarity.
Good luck with your study!

(Please note, I could not see Figure in the re-submitted documents but have read the authors description of changes.

We apologise for the confusion and the figure is now attached in the current submission.